# Metabolic Disturbances Involved in Cardiovascular Diseases: The Role of Mitochondrial Dysfunction, Altered Bioenergetics and Oxidative Stress

**DOI:** 10.3390/ijms26146791

**Published:** 2025-07-15

**Authors:** Donatella Pietrangelo, Caroline Lopa, Margherita Litterio, Maria Cotugno, Speranza Rubattu, Angela Lombardi

**Affiliations:** 1Department of Clinical and Molecular Medicine, School of Medicine and Psychology, Sapienza University of Rome, 00189 Rome, Italy; donatella.pietrangelo@uniroma1.it (D.P.); angela.lombardi@uniroma1.it (A.L.); 2IRCCS Neuromed, 86077 Pozzilli, IS, Italy; lopacaro@hotmail.it (C.L.); maggie651@hotmail.com (M.L.); maria.cotugno@neuromed.it (M.C.); 3Department of Medicine (Division of Endocrinology and Diabetes), Department of Microbiology and Immunology, Einstein-Mount Sinai Diabetes Research Center (ES-DRC), Einstein Institute for Aging Research, Fleischer Institute for Diabetes and Metabolism (FIDAM), Albert Einstein College of Medicine, New York, NY 10461, USA

**Keywords:** cardiovascular diseases, cardiac bioenergetics, mitochondria, glycolysis, metabolic flux analyzers, metabolic reprogramming

## Abstract

The study of metabolic abnormalities regarding mitochondrial respiration and energy production has significantly advanced our understanding of cell biology and molecular mechanisms underlying cardiovascular diseases (CVDs). Mitochondria provide 90% of the energy required for maintaining normal cardiac function and are central to heart bioenergetics. During the initial phase of heart failure, mitochondrial number and function progressively decline, causing a decrease in oxidative metabolism and increased glucose uptake and glycolysis, leading to ATP depletion and bioenergetic starvation, finally contributing to overt heart failure. Compromised mitochondrial bioenergetics is associated with vascular damage in hypertension, vascular remodeling in pulmonary hypertension and acute cardiovascular events. Thus, mitochondrial dysfunction, leading to impaired ATP production, excessive ROS generation, the opening of mitochondrial permeability transition pores and the activation of apoptotic and necrotic pathways, is revealed as a typical feature of common CVDs. Molecules able to positively modulate cellular metabolism by improving mitochondrial bioenergetics and energy metabolism and inhibiting oxidative stress production are expected to exert beneficial protective effects in the heart and vasculature. This review discusses recent advances in cardiovascular research through the study of cellular bioenergetics in both chronic and acute CVDs. Emerging therapeutic strategies, specifically targeting metabolic modulators, mitochondrial function and quality control, are discussed.

## 1. Introduction

Cellular metabolism encompasses the biochemical processes essential for energy production, macromolecule biosynthesis and homeostasis maintenance, serving as the fundamental engine of cellular life. Glucose, fatty acids, ketone bodies, lactate and amino acids play fundamental roles in energy generation. Glucose, the main player of the biochemical processes, is oxidized through three different subcycles: glycolysis; the Krebs cycle, also known as the TCA cycle; and mitochondrial ETC to power the synthesis of ATP [1]. In glycolysis, regardless of oxygen presence, one molecule of glucose is converted into two molecules of pyruvate, leading to ATP production and lactate formation, which acidifies the extracellular environment [1]. Changes in extracellular pH serve as an indirect indicator of the cell glycolytic capacity. In the conventional model of mitochondrial ETC, electrons derived from the TCA cycle are transferred through four protein complexes embedded in the inner mitochondrial membrane before ultimately reducing oxygen. This electron transfer is coupled with the translocation of protons into the intermembrane space, generating an electrochemical gradient across the inner membrane. This proton motive force drives ATP synthesis [2].

The study of cellular metabolism has a long history, dating back to the early 20th century. In 1920, Otto Warburg conducted pioneering research on tumor metabolism using his innovative “Warburg manometer” [3]. This device measured metabolic activity in cancer cells by detecting gas exchange through pressure changes in a system where the tissues were incubated in a flask connected to a manometer. The instrument also assessed lactate production by adding bicarbonate, thus generating carbon dioxide in proportion to the amount of lactic acid. His work was instrumental in identifying the unique metabolic pathway of cancer cells, which convert glucose into lactate even in the presence of oxygen, a phenomenon known as the “Warburg effect”. A century later, technological advancements have provided researchers with sophisticated tools to study cellular metabolism in real time. Modern instruments, such as the Seahorse Analyzer, enable precise measurement of key metabolic parameters, including OCR and ECAR [4]. The measurement of ECAR provides an indication of glycolytic activity, whereas OCR reflects the activity of oxidative phosphorylation closely tied to mitochondrial respiration.

By providing detailed insights into the metabolic states associated with various diseases, the analysis of extracellular flux assays has significantly advanced our understanding of cell biology and disease mechanisms [5]. Mitochondria, with their complex networks of functions, are essential in health regulation and represent important therapeutic targets. Dysfunction of the mitochondrial respiratory chain leads to reduced ATP production, also involving increased ROS generation, mitochondrial DNA damage and the activation of inflammatory and apoptotic pathways [6]. Bioenergetic dysfunctions are identified in many common pathologies including cancer [7]; metabolic syndrome [8]; neurodegeneration [9] and CVDs [10]. Specifically, the study of energy metabolism is particularly critical in CVDs due to the heart’s high energy consumption and its consequent dependence on the correct functioning of mitochondria. Mitochondrial dysfunction is indeed detected in association with all major cardiac diseases and with the final stage of HF.

Since mitochondrial targets are becoming increasingly pivotal for the treatment of heart-related diseases, the objective of this review is to discuss the most relevant recent findings regarding cardiac and vascular bioenergetics in major common CVDs, providing novel insights into disease pathogenesis and new targets for therapeutic strategies. In particular, the study of metabolic flux assays in CVDs is mainly used to explore the metabolism of cardiomyocytes, endothelial cells, vascular smooth muscle cells and neurons. In this context, the analysis of cellular bioenergetics allows researchers to monitor the effectiveness of new therapeutic strategies aimed at improving mitochondrial function and energy metabolism within the heart and vasculature.

## 2. Cellular Metabolism in Cardiovascular Health and Disease

The heart beats more than 3 billion times in an average human life. However, despite hydrolyzing 20 times its mass in ATP each day, it only stores enough energy for several heartbeats [11]. For this reason, the heart must adapt to changing nutritional supply and physiological demands to meet ongoing energy needs. Energy production within the heart is mainly via mitochondrial oxidative phosphorylation and glycolysis of substrates such as FA, lactate, glucose, ketones and amino acids [12,13]. Upon entry into mitochondria, oxidation of the different substrates convergently leads to acetyl–CoA production and entry into the TCA cycle, which subsequently generates ATP via oxidative phosphorylation. A series of transporters and enzymes need to be in place to ensure the timely uptake of substrates from the bloodstream into cardiomyocytes and the transport through the mitochondrial membrane inside the cells. In the healthy heart, these processes are intricately regulated by the balance between intermediates (e.g., FAD/FADH2 and NAD +/NADH in the mitochondrial matrix, acetyl–CoA/CoA, ATP/AMP in cardiomyocytes) that are reflective of the current energy status. Under diseased conditions, dysregulation of the key players in these processes leads to inefficient cardiac energy production [14]. Hence, changes in levels of the long-chain FA mitochondrial carrier were found to be associated with increased risk of HF [15,16]. Similarly, changes in blood sugar levels resulting from switching from FA metabolism to carbohydrate metabolism and ketone oxidation, as well as increased plasma levels of branched-chain amino acids resulting from inefficient oxidation, also reflect disruption in heart function [17].

CVDs refer to diseases that are related to the heart and the vascular system supporting it. With the growing appreciation of the metabolic basis of CVDs and being aware that most of the CVD risk factors have their roots in metabolic disruptions, the study of cellular bioenergetics through metabolism analyzers has become a powerful investigative tool for various aspects of cardiac diseases.

## 3. Essential Hypertension

Hypertension is the leading cause of death globally [18], affecting more than 1.3 billion individuals worldwide, with a higher prevalence in men [19]. The pathogenesis of hypertension is intimately tied to dietary salt intake. Salt sensitivity is influenced by multiple factors, including genetic predisposition, ethnicity, age, sex, body mass index and coexisting conditions such as diabetes, kidney disease, and metabolic syndrome [20]. Approximately 50% of patients with hypertension have SS-HTN with salt intake dependent-BP changes [20]. The first signs of SS-HTN-dependent renal disease are high proteinuria levels, glomerular damage and loss of podocytes. The reason for these abnormalities can be found in functionally and structurally defective glomerular mitochondria. Of interest, glomeruli from rats fed with a high-salt diet showed overproduction of ROS, as compared to the control group, caused by structurally damaged mitochondria [21].

An important contribution to the development of hypertension is given by the RAAS, where Ang II is the main actor. Ang II is strongly involved in renal damage by inducing renal vascular dysfunction, inflammation and oxidative stress [22]. Overproduction of Ang II has several effects on vascular ECs, including the release of endothelium-derived vasoconstrictors and, in contrast, the reduction of endothelium-derived vasodilators such as NO. In ECs, mitochondrial oxidative stress, induced by Ang II overproduction, contributes to lower NO levels and favors endothelial dysfunction. These phenomena result in endothelial damage, as well as water and sodium retention [22]. Ang II also reduces SIRT3 and SIRT6 transcription level in ECs, decreasing SIRT3 localization in mitochondria and generating mt-ROS [23]. Moreover, ATP is synthetized by dysfunctional ECs through glycolysis, thereby preserving their mitochondrial membrane potential. Furthermore, autophagy may be activated to maintain glycolytic-dependent ATP production [24].

In the experimental model of RRAEC dysfunction, Ang II upregulated glycolysis and promoted autophagy [24]. Differential expression profiles of potential candidate genes involved in the Ang II effects in RRAECs led to the discovery of several microRNAs mediating the Ang II-induced RRAEC injury by affecting glycolysis levels, mechanisms involved in cell metabolism and differentiation, and mechanisms involved in autophagy and repair through the activation of mTOR signaling [24].

Multiple evidence supports the role of low levels of ANP in the development of SS-HTN. ANP, encoded by *Nppa*, regulates BP, vasodilation and renal salt excretion through the activation of the NPR-A receptor and cGMP/PKG signaling [25]. Moreover, ANP/cGMP/PKG signaling plays a protective role by improving mitochondrial respiration and oxidative capacity in mice and human skeletal muscle [26,27,28], by enhancing mitochondrial biogenesis and function, finally contributing to improved oxidative metabolism and reduced ROS production [26,27,28]. Additionally, exogenous ANP administration attenuated Ang II-stimulated oxygen consumption in renal mitochondria, contributing to the preservation of mitochondrial efficiency and mitochondrial biogenesis [29]. Of note, the impact of ANP deficiency on mitochondrial bioenergetics was tested in the kidney of a hypertensive animal model, Dahl salt-sensitive *Nppa*-knockout rats (SS^NPPA−/−^) fed with a high-salt diet. This model displayed mitochondrial damage, fragmentation, decreased membrane potential and higher levels of mitochondrial ROS compared to Dahl SS wild type rats (SS^WT^). Interestingly, renal mitochondria from SS^NPPA−/−^ rats showed increased biogenesis and turnover of damaged organelles as compensatory responses. Consistently with this adaptation, an increase in mitochondrial respiration and OCR levels were detected in SS^NPPA−/−^ compared to SS^WT^ rats. Therefore, ANP deficiency leads to compromised mitochondrial bioenergetics and dynamics in the kidney that may worsen hypertension and the related renal damage [30]. These discoveries emphasize the essential role of ANP in preserving mitochondrial integrity and bioenergetic homeostasis. In humans, low levels of ANP, associated with a specific gene promoter variant or with metabolic abnormalities such as obesity and metabolic syndrome, significantly increase the predisposition to develop hypertension [25,31]. Based on this evidence, either the stimulation of ANP signaling or a pharmacological increase in ANP levels could represent an efficacious therapeutic strategy for mitigating hypertension-mediated renal damage.

Hypertension also has an important role in metabolic syndrome, characterized, as defined by the International Diabetes Federation, by high BP, dyslipidemia (with increased triglycerides and reduced high-density lipoprotein cholesterol), increased fasting glucose and central obesity [32]. In this context, a role for SCFAs has been uncovered [33]. These gut microbiota products exert several positive functions within the cardiovascular system, such as BP regulation, improvement of microcirculation and anti-inflammatory effects [34]. They also improve insulin sensitivity and lipid profiles while reducing body weight in metabolic syndrome. Therefore, SCFAs become a key player in hypertension as well as in other CVDs.

Preeclampsia represents a multisystemic disorder associated with hypertension and vascular disfunction during pregnancy, causing maternal and fetal mortality. sFLT-1 and Ang II play a pivotal role in preeclampsia, leading to higher oxidative stress and mitochondrial dysfunction. The effects of sFLT-1 are inhibited by CALCA, a vasodilator that counteracts Ang II-dependent increased BP and mitochondrial oxidative stress. The inhibition of sFTL-1 via CALCA was evaluated in the omental artery smooth muscle cells, where OCR was enhanced and oxidative stress reduced after CALCA treatment. Mitochondrial function was also restored in the presence of CALCA, decreasing mitochondrial fragmentation and increasing the levels of genes involved in mitophagy such as DRAM1 and PINK1. Finally, several studies performed in preeclampsia mouse models showed that BP and Ang II-sensitivity were reduced after CALCA administration [35].

## 4. Pulmonary Hypertension

PH is a consequence of increased pulmonary vascular resistance and has a prevalence of ~1% in the global population, being higher in elderly individuals [36]. Mounting evidence indicates that inflammation is a core component of PH. In fact, prevailing features of PH include sustained inflammation around the pulmonary blood vessels and vascular remodeling, eventually leading to enlargement of the right ventricle and potentially fatal outcomes [37,38,39]. Some cellular alterations associated with this pathology are PASMC proliferation, mitochondrial dysfunction and apoptosis resistance, all of which play a pivotal role in pulmonary arterial remodeling. Abnormal PASMC proliferation, together with PAEC dysfunction and excessive extracellular matrix deposition, contributes to increased pulmonary arteriolar tone due to vasoconstriction, as well as to pulmonary vascular wall thickening, remodeling, fibrosis and inflammation (Figure 1).

These structural vascular changes increase pulmonary vascular resistance and pulmonary arterial pressure, ultimately leading to right ventricle overload. Therefore, acting on PSMC proliferation may represent a therapeutic approach to restore pulmonary vascular homeostasis and slow PH clinical progression [40]. One of the key mechanisms by which PH-PASMCs overpass apoptosis and display their hyperproliferative phenotype is the metabolic shift toward glycolysis for ATP production. This metabolic reprogramming reduces PDH activity and promotes apoptosis resistance, aggravating pulmonary vascular remodeling and PH progression [39]. Indeed, experimental evidence shows that lower PDH activity promotes PASMC proliferation, resistance to apoptosis and PH progression [41]. A strategy to prevent this process is to improve PDH activity to reduce uncontrolled apoptosis and counteract PH progression. The inhibition of FA oxidation is a way to achieve PDH activation, contributing to reverse metabolic shift in PH PASMCs [42]. To achieve this goal, recent studies used FPER-1 to inhibit CPT-1, the enzyme that controls long-chain FA access in β-oxidation mitochondria site, reducing their uptake in mitochondria [43]. Griffiths and coll. used the fluorinated derivative of FPER-1 to investigate the effects on PDH activity in PASaMCs from PH patients [44]. The treatment of FPER-1 successfully attenuated cell proliferation in PASMCs, suppressing the PI3K/Akt axis, a pathway strongly involved in cell proliferation in PH. Moreover, FPER-1 induced PDH activation, decreased glycolysis and improved mitochondrial respiration, thus reducing the Warburg effect. Overall, these results highlight the role of FPER-1 as a potential novel therapeutic drug to counteract PASMC proliferation in PH by ameliorating vascular remodeling [44]. Of interest, Ma et al. investigated the role of AIF, a mitochondrial oxidoreductase, in the regulation of mitochondrial energy metabolism and mitophagy in PASMCs [45]. Under hypoxia, the expression of AIF is reduced in both in vivo and in vitro models of PH. This condition leads to a defective mitochondrial respiratory chain, reduced oxidative phosphorylation and higher glycolysis with consequent higher ROS production. Also, mitophagy and autophagy are significantly compromised due to AIF deficiency. These data suggest the importance of AIF as a key factor to regulate post-transcriptional mechanisms during mitochondrial dysfunction induced by hypoxia in PASMCs [45].

The effects of FAS on mitochondrial function were reported in both in vitro and in vivo mouse models of hypoxia-induced PH [46]. In this study, FAS expression was significantly increased in lungs isolated from hypoxia-induced PH mice and in human PASMCs. Notably, FAS pharmacological inhibition by a specific molecule called C75 improved mitochondrial function and right ventricle function in the animal model. At the same time, in vitro FAS inhibition by C75 improved mitochondrial respiratory capacity, increased ATP levels and decreased oxidative stress levels. In summary, these data point out FAS inhibition as a tool to successfully treat PH in humans. Among the various mechanisms involved in PH, metabolic reprogramming is one of the initiating processes where mitochondrial network remodeling plays a crucial role [46]. Indeed, the aim of a recent study by Yegambaram and coll. was to evaluate changes in mitochondrial dynamics before the manifestation of PH. Of note, an early-stage PH model showed higher levels of mitochondrial fusion proteins Mfn1, Mfn2 and Opa1. Specifically, the overexpression of Mfn1 in PECs caused metabolic reprogramming by disrupting the activities of complexes I and III within the ETC, ultimately increasing mitochondrial ROS levels. These alterations caused an enhanced glycolysis, supporting the evidence that mitochondrial fusion and Warburg phenotype are strongly associated [47].

Finally, the role of the glycolytic protein ENO1 has been extensively investigated in the pathogenesis of PH. In their study, Shi et al. revealed that ENO1 expression was upregulated in lung tissue of patients with PH and in both in vitro and in vivo models of hypoxic PH. In vitro, the overexpression of ENO1 enhanced hypoxia-related endothelial dysfunction in human PAECs, whereas inhibition of ENO1 reversed this condition. Similar results were shown in vivo, where mice treated with a specific ENO1 inhibitor exhibited an improvement in right ventricular function and reduced PH progression. Hence, targeting ENO1 could be a strategy to ameliorate PH by improving endothelial and mitochondrial dysfunction through the PI3K-Akt-mTOR pathway [48]. A recent study has demonstrated the correlation between SOX17 gene mutations and PH [49]. In particular, the risk of developing PH is correlated with SOX17 deficiency. Desai and coll. hypothesized that SOX17 could improve mitochondrial function and alleviate PH progression by inhibiting HIF2α. In the same study, oxidative phosphorylation and mitochondrial function were significantly promoted when SOX17 was overexpressed in PAECs [49].

## 5. Heart Failure

Under normal fasting conditions, the healthy heart generates 60–90% of its energy through the mitochondrial oxidation of FA, with the remainder supplied by the oxidation of glucose, lactate and ketones [50]. Pathological structural remodeling such as cardiac hypertrophy results in well-described reprogramming of cardiac metabolic pathways, leading to an overall increased reliance on glucose metabolism (increased glucose uptake and glycolysis) with a decrease in FA oxidation (Figure 2).

These metabolic changes represent an early event in HF pathogenesis [51]. Indeed, prolonged reliance on glucose likely results in an ultimate state of ATP depletion and bioenergetic starvation, as well as altered cardiac contractility and function leading to HF [52]. Moreover, during the initial phases of HF, mitochondrial number and function progressively decline, causing an overall decrease in the oxidative metabolism of most fuels and a propagation of energy deficit [53]. In this context, inhibition of CPT-1, the enzyme regulating mitochondrial FA oxidation, prevents LV wall thinning and delays progression to end-stage disease [54].

In 2023, Li et al. discovered that disabling FA oxidation in cardiomyocytes improves resistance to hypoxia and stimulates cardiomyocyte proliferation, allowing cardiac regeneration after ischemia–reperfusion injury [55].

Furthermore, since end-stage HF, whether of ischemic or non-ischemic origin, presents a similar phenotype of myocardial hibernation [56], it is plausible to hypothesize that myocardial hibernation represents a common hallmark resulting from an adaptive metabolic response following the shift from FA oxidation to glucose metabolism.

Endothelial bioenergetics plays a critical role in HF, particularly in HFpEF. Some evidence indicates that SIRT3 deficiency in Ecs leads to a significant reduction in glycolysis, increased mitochondrial respiration and a more pronounced production of ROS. SIRT3 deficiency has also been associated with marked increases in p53 acetylation, endothelial apoptosis and senescence. The deterioration of endothelial metabolism mediated by SIRT3 may lead to impaired communication among Ecs, pericytes and cardiomyocytes, as well as to coronary microvascular rarefaction. This, in turn, promotes cardiomyocyte hypoxia, titin-based cardiomyocyte stiffness and myocardial fibrosis, ultimately leading to diastolic dysfunction and HfpEF [57,58].

In recent decades, research on the metabolic changes that occur during HF has primarily focused on changes in FA and glucose metabolism. A recent study demonstrated that inhibition of 12(S)-HETE, a marker elevated in plasma of individuals and mice with diabetes, restored mitochondrial function in the endothelium, improving the ability of blood vessels to dilate in HfpEF patients. Moreover, their data suggest that modulators of TRPV1, a receptor activated by 12(S)-HETE, could exert therapeutic potential for treating vascular dysfunction and HfpEF in patients with diabetes and obesity [59]. These studies are very relevant considering that HfpEF represents up to 50% of HF cases in the United States [60].

Mitochondrial alterations play a key role in the development of DCM. In a study by Fiordelisi and coll. [61], the effect of arginine supplementation was evaluated on cardiac mitochondrial function and DCM development in db/db mice, a known model of T2D. Indeed, in these animals, arginine supplementation preserved diastolic function and LV morphology, and it improved exercise tolerance and the propensity to physical activity. Mitostress test experiments performed in cardiomyocytes isolated from db/db mice showed that arginine supplementation increased mitochondrial respiration and mitochondrial biogenesis [61]. These improvements were also associated with significant reduction of miR-143, an index of mitochondrial damage. Moreover, CCR2+ monocytes and macrophages isolated from the peripheral blood of DCM patients exhibit metabolic reprogramming leading to alterations of their inflammatory phenotype.

The inflammatory metabolism in CCR2+ macrophages and monocytes promotes a state of chronic inflammation through increased glycolysis, production of ROS and release of cytokines harmful to myocardial tissue, such as IL-1β, IL-6 and TNF-α. These cytokines impair excitation–contraction coupling, promote fibroblast activation and extracellular matrix remodeling (via TGF-β, MMPs) and induce apoptosis in cardiomyocytes. As a result, impaired contractile function and ventricular dilatation are observed [62]. Interestingly, the suppression of GLUT1 reduced the amount of mtROS and the expression of inflammatory factors, thereby limiting the progression of DCM [62]. In addition, it has been discovered that BDH1 may be an early biomarker of metabolic stress that precedes DCM [63]. However, whether the increased reliance on ketones for energy is an adaptive or maladaptive process in the failing heart remains a matter of debate [64,65,66]. Another model used to study DCM is the SOD2 knockout mouse. Lack of SOD2 leads to increased ROS and overproduction of 4-Hydroxynonenal in mitochondria, causing mitochondrial dysfunction. In this model, a metabolic shift occurs with a preference for glucose over FA [67].

Preliminary results suggest that Salsolinol can be useful in HF. Salsolinol is a plant-based isoquinoline alkaloid, and it has been tested in a doxorubicin-induced chronic HF rat model and in H9c2 cardiomyocytes. This molecule improves mitochondrial respiratory function and energy metabolism by inhibiting the excessive activation of a specific mitochondrial calcium uniporter in H9c2 cardiomyocytes, thereby promoting ATP production [68]. Of interest, the same beneficial effects were found with Higenamine, a natural benzylisoquinoline alkaloid isolated from Aconitum, combined with (6)-gingerol [69].

Perhaps not surprisingly, mitochondrial dysfunction, altered morphology, and reduced contractility were observed under iron deficiency conditions, which are common in HF patients. Additionally, there is reduced activity of Fe-S cluster-based complexes in the mitochondria of human cardiomyocytes, along with a metabolic shift from FA oxidation to anaerobic glycolysis [70]. These effects can be reversed by iron supplementation. Similar results were achieved in rats with CKD that are prone to develop HF. In this animal model, a single-dose parenteral iron therapy ameliorated oxidative stress in cardiac tissue and led to an improvement of cardiac function. During the early stages of CKD, this strategy could reduce vulnerability to oxidative stress and mitigate the adverse cardiovascular outcomes associated with the renal disease [71].

Extracellular flux assay approaches may be particularly useful to investigate the potential mechanisms underlying the cardioprotective effects of sodium–glucose cotransporter 2-inhibitors (SGLT2i) [72]. The results of a trial called the Dapagliflozin (a specific SGLT-2i)-HF trial [73,74] were among the first to imply that the advantages observed in HF cannot be attributed solely to the blood glucose-lowering effects. Among the primary pathways thought to be involved in SGLT-2i actions, an improvement in cardiac mitochondrial bioenergetics with a reduction in ROS production is thought to play a role. SGLT-2i mechanisms of action have been linked to a moderate but significant increase of circulating ketone bodies, which have been shown to play a positive adaptive role in HF [75,76,77,78]. We and others have demonstrated that empagliflozin, another specific SGLT-2i, significantly reduces mitochondrial calcium overload in the human vascular endothelium. Of note, ROS production triggered by high glucose in endothelial cells is ameliorated by empagliflozin, improving cell viability in response to oxidative stress [79,80].

## 6. Atherosclerosis

Atherosclerosis is characterized by EC dysfunction, which promotes the recruitment and accumulation of monocytes. They differentiate into macrophages and internalize lipids, transforming into cholesterol-laden foam cells. Furthermore, dysfunctional Ecs produce ROS and promote vascular inflammation and thrombosis, while reducing vasodilatation [81] (Figure 3).

Macrophages are key pathological elements in the progression of atherosclerosis and of its thrombotic complications. The advancement of atherosclerotic plaques is sustained by a continuous recruitment of monocytes from the bloodstream, which contribute to the progressive accumulation of macrophages within the lesion. In response to pathological conditions such as hypertension, these macrophages release several pro-inflammatory cytokines that promote both the formation and progression of atherosclerotic plaque [82]. Macrophages also play a critical role in plaque destabilization by promoting cell death and necrosis of the plaque core, thereby facilitating rupture and the onset of acute thrombotic events in affected arteries [83]. Under conditions of high oxidative stress, macrophage death has been observed in both in vivo and in vitro models, often accompanied by increased lipid accumulation. In particular, lipid content enriched in leucine within macrophages may stimulate their activation and contribute to plaque rupture [84].

Our understanding of atherosclerosis pathogenesis has improved in recent years, challenging many previous notions [85]. Coronary atherosclerosis now affects an increasing number of younger people and more women. Furthermore, even some non-traditional factors, such as disturbed sleep, physical inactivity, the microbiome, air pollution and environmental stress are emerging as causative factors of atherosclerosis [86]. High cholesterol remains one of the most important predisposing factors for the development of atherosclerosis. Both crystalline and soluble forms of cholesterol are known to increase macrophage secretion of pro-inflammatory interleukins, thereby exacerbating the inflammatory response in atherosclerosis. Using the Seahorse platform, it was demonstrated that vascular cells treated with cholesterol for 16 h exhibited a decline in mitochondrial function, including the basal oxygen consumption rate, ATP production, maximal respiratory capacity and spare respiratory capacity. The study revealed that MitoTEMPO, when combined with cholesterol, restored all parameters of mitochondrial function, highlighting its potential as an anti-atherogenic drug [87]. Interestingly, MitoTEMPO inhibits mtROS production and suppresses endothelial cell activation and aortic monocyte recruitment in ApoE knockout mice. This suggests that mtROS mediate lysophosphatidylcholine-induced endothelial activation in the early phase of atherosclerosis [88].

LDL and HDL are well-established cardiovascular risk factors [89]. During the early stages of atherogenesis, LDL and other lipoproteins accumulate in the arterial wall [90]. Under conditions of oxidative stress, oxLDL is formed, which promotes inflammation and increases vascular permeability, thereby driving the progression of atherosclerosis [91]. Atherogenic LDL particles are enriched in phospholipids and sphingolipids, such as sphingomyelin and ceramide. Both HDL and LDL, through their interaction with sphingolipids, influence EC metabolism primarily by increasing ROS production and promoting eNOS uncoupling. Consequently, these lipoproteins can be considered potential therapeutic targets in both the early and advanced stages of atherosclerosis [92,93].

Several investigators used Seahorse technology as a successful approach to address the contribution of LDL in atherosclerosis. A study by Braczko et al. demonstrated that mild chronic dyslipidemia caused energy and metabolic alterations before affecting the mechanical function of the heart in LDLR knock-out mice [94]. Notably, there are various forms of LDL. Among them, L5 is particularly atherogenic since it induces mitochondrial dysfunction through the opening of the mPTP and by altering mitochondrial fission. It has been discovered that ApoE may play the most important role in L5-induced mitochondrial dysfunction in cardiomyocytes, and it does it via its interaction with the voltage-dependent anion-selective channel [95]. Mitochondrial metabolism is compromised in an ApoE^−/−^/LDLR^−/−^ murine model. Therefore, the metabolism of vascular cells tends to shift towards glycolysis, the PPP and the hexosamine biosynthetic pathway, which are all associated with increased inflammation and cellular stress. In this context, mitochondrial metabolism was restored by blocking the ALOX12 pathway and activating AMPK signaling [96]. Moreover, an ApoE^−/−^ mouse model was used to evaluate the beneficial effects of a mitochondrial protective agent, a molecule called AR-C17. This compound may serve as a promising grain-based dietary bioactive ingredient for the prevention of atherosclerosis, also supporting the proposed health claims associated with a whole-grain diet [97].

Moreover, the protein KLF7 appears to reduce atherosclerotic lesions by limiting glucose metabolic reprogramming in macrophages from mice undergoing high-fat dietary feeding. By applying the Seahorse approach, it was reported that the pathways underlying the protective effect of KLF7 involve the activation of HDAC4 which inhibits miR-148b-3p and promotes the transcription of NCOR1 [98].

Triglycerides contribute to atherosclerosis development. The effects of postprandial TGRL exposure on brain microvascular endothelial cells were reported in a study by Nyunt et al. [99]. The authors showed that TGRL lipolysis products increased ROS production, causing mitochondrial stress. The significance of this study lies in the discovery of two variants of ATF3, with the ATF3-T4 variant behaving as the most important in regulating inflammation induced by lipolysis products [99].

Finally, exposure to high glucose levels causes mitochondrial damage. Of interest, it has been demonstrated that liraglutide, used primarily to treat T2D, improves endothelial cell function by downregulating PINK1/Parkin-mediated mitophagy and increasing eNOS phosphorylation [100].

Overall, diabetes accelerates atherosclerosis through mechanisms such as hyperglycemia, oxidative stress, chronic inflammation, and epigenetic dysregulation, ultimately promoting the formation of unstable plaques. Oxidative stress plays a central role in the pathogenesis of atherosclerosis associated with diabetes [101,102,103].

In Ecs, hyperglycemia leads to increased mitochondrial ROS production, primarily through the activation of NOX proteins [104]. Among these, NOX1 plays a pivotal role in diabetes-induced atherosclerosis. In Apoe+ mouse models with induced diabetes, either genetic deletion of *Nox1* or pharmacological inhibition with GKT137831 (setanaxib, a dual NOX1/NOX4 inhibitor) result in a significant reduction in the atherosclerotic burden [105].

Finally, an important molecular player in cardiovascular risk is represented by JAK2 kinase. In myeoloproliferative neoplasms, a gene mutation, the AK2-V617F mutation, has been found to be significantly associated with chronic inflammation, increased cytokines release, oxidative stress, atherosclerosis and thrombotic events [106]. Moreover, pulmonary hypertension and heart failure add to the cardiovascular burden of gene mutation carriers. This evidence strongly underlies the tight link between vascular inflammation and its consequences.

## 7. Myocardial Infarction

AMI is a leading cause of morbidity and mortality worldwide [107]. AMI occurs more frequently in males and in individuals over the age of 65 years. Among the comorbid conditions associated with AMI, hyperlipidaemia is the most prevalent one, followed by hypertension, DM and NASH [108]. A contribution to AMI occurrence is given by obesity and DM, independently of hypertension and CAD [109]. AMI shares several pathophysiological features with other metabolic heart diseases, including mitochondrial dysfunction, nitro-oxidative stress, impaired cellular metabolism and chronic inflammation. These alterations compromise myocardial structure and energetics, leading to subclinical myocardial infarction and to HF development [110]. Moreover, these biochemical alterations worsen myocardial susceptibility to ischemic injury, along with the reperfusion procedure following AMI, which leads to a second type of damage known as the I/R injury [111].

During cardiac I/R injury, the impairment of endothelial barrier function, increased endothelial permeability and significant cellular swelling promote microthrombus formation and obstruction of small vessels [112]. Additionally, mitochondrial dynamic is altered, being characterized by increased fission, decreased fusion, and disrupted mitophagy [113].

Various mechanisms contribute to the production of ROS, particularly mtROS. Oxidative stress induced by mtROS triggers endothelial apoptosis, pro-inflammatory responses, alterations in glucose metabolism, capillary tube formation, endoplasmic reticulum stress, intracellular calcium imbalances, and platelet activation (with P-selectin expression and αMβ2 integrin activity) [114].

In addition, mtROS regulate multiple signaling pathways in Ecs during cardiac I/R injury. Firstly, they reduce the expression of eNOS, which can lead to vasospasms [115], thereby limiting blood flow to the reperfused heart. mtROS-induced oxidative stress accelerates EC senescence, characterized by impaired migratory response, reduced paracrine capacity and increased endothelial permeability. Finally, chronic mtROS accumulation contributes to vascular fibrosis and remodeling, secondary to endothelial apoptosis, representing a risk factor for arterial stiffness [116].

Mitochondrial dynamics, which regulate morphologically and functionally mitochondrial homeostasis, is controlled by a set of GTPases including Mfn1/Mfn2, Opa1 and Drp1, involved in mitochondrial fusion and fission, respectively. Disruption of mitochondrial dynamics leads to mitochondrial dysfunction increasing oxidative stress, thereby contributing to myocardial infarction [117]. Enhanced mitochondrial fission and reduced fusion result in mitochondrial fragmentation, excessive ROS production and impaired ATP production. Ischemia-induced fission is primarily driven by Drp1 translocation to the mitochondria outer membrane, triggered by intracellular Ca2+ overload and ROS accumulation. Moreover, reperfusion injury further activates Drp1 and its receptors, exacerbating mitochondrial fission and consequently mitochondrial fragmentation. In contrast, impaired mitochondrial fusion, due to Mfn1/Mfn2 downregulation and decreased Opa1 levels, promotes mitochondrial fragmentation independently from Drp1 [117,118].

The study of metabolic flux alterations involved in AMI, I/R injury and post-MI ventricular remodeling helped to dissect some of the contributory molecular mechanisms and to investigate new therapeutic approaches for these conditions. In this context, it has been reported that metformin exerts a cardioprotective effect by modulating both the complex I activity of the ETC and the mPTP opening [119]. The in vivo acute administration of a high dose of metformin rescued the cardiac dysfunction induced by the ischemic insult. In vitro, cardiomyocytes under I/R condition exposed to metformin preserved both mitochondrial function and integrity as measured by the Seahorse OCR, which showed a rescued basal and maximal respiration rate. In addition, metformin rescued both impaired cardiomyocyte contractile function and calcium signaling [120].

Cardiac repair after MI consists of three phases: the inflammatory phase, the reparative–proliferative phase, and the maturation phase [121]. Following AMI, a significant proportion of peripheral monocytes migrates to the ischemic region and differentiates into macrophages. Ly6C^high^ monocytes differentiate into pro-inflammatory macrophages, stimulating the inflammatory response. These macrophages are characterized by a predominantly glycolytic metabolism with reduced activity of the TCA cycle and OXPHOS. This rapid anaerobic metabolism supports their function in acute inflammatory response, promoting the production of nitric oxide and pro-inflammatory cytokines [122]. Whereas Ly6C^low^ monocytes differentiate into pro-reparative macrophages, they exhibit a more oxidative metabolism, with enhanced activation of OXPHOS, fatty acid metabolism, and glutaminolysis. This metabolic profile promotes the production of anti-inflammatory factors and supports tissue repair, including extracellular matrix remodeling and angiogenesis [121].

Immunometabolic plasticity represents a key driver in the transition between macrophage phenotypes, as cellular metabolism regulation is not merely a consequence of differentiation but an active mechanism that modulates and directs the pro-inflammatory or pro-reparative functions of macrophages [123].

In the early phase after MI (days 1–3), ischemia and hypoxia lead to a reduction in OXPHOS, making energy production largely dependent on glycolysis and the PPP. Two breakpoints are observed in the TCA: the first at IDH and the second at SDH. Additionally, damage to the SDH enzymatic complex inhibits the conversion of succinate into fumarate, leading to succinate accumulation. The latter can stabilize HIF1α, inducing glycolytic reprogramming and the transcription of pro-inflammatory genes [124].

In a study by Zhang et al., it was found that the lack of NPM1 enhances the reparative function of cardiac macrophages by shifting their metabolism from an inflammatory glycolytic system to oxygen-based mitochondrial energy production. As a result, the administration of an antisense oligonucleotide targeting NPM1 or the oligomerization inhibitor NSC348884 reduces tissue damage and promotes cardiac healing post infarction in mice [125].

Short-term treatment with AOAA also improves cardiac function post MI in mice by inhibiting the activation of the inflammatory NLRP3-Caspase1/IL-1β pathway and reducing the release of pro-inflammatory cytokines IL-6 and TNF-α, while increasing the level of the anti-inflammatory cytokine IL-10 in both the ischemic myocardium and M1 macrophages [126].

Muscone, an active monomer belonging to musk, positively modulates inflammation, angiogenesis, myocardial remodeling and cardiac function related to AMI [127,128,129]. In vitro experiments in AC16 cells exposed to I/R confirmed the known anti-inflammatory activity of muscone along with a significant anti-apoptotic property. Of note, muscone promoted the glycolytic flux of AC16 cells after I/R by upregulating PKM2 levels through increased expression of H3K4me3 and reduced expression of both H3K27me3 and H2AK119Ub in the PKM2 promoter region [130].

DMF acts as a protective agent in myocardial remodeling. In fact, DMF reduced LV infarct area and LV dilation [131]. An improvement in collagen deposition, myofibroblast activation and angiogenesis was also observed after 7 days of DMF treatment. In vitro, DMF attenuated the inflammatory pathway and enhanced the anti-inflammatory response and the pro-reparative phenotype in macrophages and fibroblasts isolated from an infarcted heart [131]. The Seahorse analysis revealed that DMF differently affected macrophage and fibroblast metabolism, by improving oxidative phosphorylation levels in the former while decreasing it in the latter. Thus, DMF modulated the mechanisms underlying post-MI myocardial remodeling through distinct metabolic effects in fibroblasts and macrophages [131].

MicroRNAs are noncoding RNAs that target mRNAs to prevent their translation. The literature reports the involvement of few microRNAs in AMI and subsequent ventricular remodeling. MicroRNA 210 promotes the metabolic shift from oxidative phosphorylation to glycolysis as a compensatory protective mechanism of the heart during AMI. It regulates glycerol-3-phosphate dehydrogenase, mitochondrial energy metabolism and ROS flux, finally improving cardiac function. MitoQ, a mitochondria-targeted antioxidant, was able to compensate the effect of miR-210 deficiency on mitochondrial respiration and to improve significantly the coupling efficiency of energy production in the heart of miR-210 KO mice [132]. The transient ectopic expression of miR-294, physiologically expressed only during the development and prenatal stages, restored typical developmental signals in cardiomyocytes promoting cell cycle reactivation, increasing oxidative phosphorylation and glycolysis and reducing apoptosis. When miR-294 was administered to mice for 14 days following AMI, it improved LV function and reduced the infarct size [133].

The increased expression of miR-214–3p is associated with cardiac remodeling in the acute phase after experimental AMI. It was observed that miR-214–3p increased the expression of the lysyl oxidase protein family, suggesting a key role in extracellular matrix turnover and fibrosis. Moreover, the overexpression of miR-214–3p led to decreased mitochondrial activity and density, as well as to decreased expression of an array of mitochondrial proteins, including mitofusin-2, ultimately resulting in mitochondrial dysfunction [134].

By targeting the above discussed microRNAs, novel therapeutic strategies for AMI and ventricular remodeling could be developed.

## 8. Stroke

Stroke is the second most common cause of death and disability worldwide [107]. Atherosclerosis and acute vascular occlusion cause ischemic stroke (IS). Therapeutic thrombolysis is the only effective treatment that re-establishes blood flow in the ischemic brain. Sudden decreases in oxygen and nutrient supply lead to changes in tissue metabolism, particularly in FA and glucose oxidation [135]. Moreover, the sudden reintroduction of oxygen and nutrients during reperfusion leads to I/R injury, triggering oxidative damage and ultimately cellular death. This condition, caused by the accumulation of free radicals, significantly increases the inflammatory response and BBB damage [136]. One of the known pathological mechanisms underlying I/R injury is the production of ROS, with the mitochondrial respiratory chain being the principal contributor, compromising mitochondrial function and energy production [137]. Beyond ischemia, oxygen and glucose deprivation also lead to mitochondrial dysfunction and increased ROS generation, which is exacerbated during reperfusion, where oxygen is used as a substrate for ROS production. The overproduction of ROS leads to opening of the mPTP and mitochondrial membrane depolarization that induces the activation of pro-apoptotic signaling and mitochondrial-dependent apoptosis. The pathophysiological outcomes of stroke are the results of these mitochondrial alterations. ROS overproduction damages ECs, disrupting tight junctions, increasing BBB permeability and causing cerebral oedema. Endothelial injury and inflammation facilitate the formation of microthrombi, impairing cerebral microcirculation and contributing to reperfusion injury and ischemic lesions. For these reasons, preventing mtROS production and activating antioxidant defense may be a strategy to counteract the consequences of I/R injury [138].

Of note, FtMt is a mitochondrial iron storage protein known to play an antioxidant role in neurodegenerative diseases by chelating free iron. In mitochondria, where ROS generation is enhanced by the presence of free iron, FtMt protects against oxidative damage [139]. Wang and coll. found that FtMt exerts a protective effect against neuronal apoptosis in I/R-exposed brains and positively regulates mitochondrial bioenergetics and glucose metabolism [140]. The data suggest that FtMt is upregulated in I/R brains and that deficiency of this protein causes brain damage and neurological deficit in I/R-exposed mice. Moreover, FtMt deficiency promotes neuronal apoptosis in I/R-exposed brains through the activation of both mitochondrial and ER stress [140]. In contrast, FtMt overexpression suppresses the apoptosis induced by OGD/R and ER activation, contributing to the restoration of mitochondrial iron overload, mitochondrial function and ROS reduction. Both increased ATP content and respiratory capacity in the presence of FtMt were detected under I/R conditions, indicating protection against OGD/R energy stress. Moreover, neuronal cells overexpressing FtMt showed higher glucose consumption and increased production of NADPH and glucose-6-phosphate dehydrogenase after OGD/R insult, displaying activation of the pentose–phosphate pathway to maintain energy homeostasis. Thus, FtMt protects against I/R-dependent apoptosis by improving mitochondrial bioenergetics and energy metabolism and limiting ROS production [141].

Lipids play a crucial role in IS. Phosphatidylcholine acyl remodeling, sphingolipid metabolism and free FA–lipid droplets are involved in stroke pathogenesis, affecting the interplay between neurons and astrocytes [142]. These alterations cause neuronal accumulation of long-chain acylcarnitines (LCACs), intermediates of FA oxidation, and their transport through the mitochondrial membrane. LCACs increase in the peripheral blood of stroke patients and in mice models of transient middle cerebral artery occlusion. In this regard, it has been shown that increased levels of LCACs accumulate in mitochondria of OGD/R astrocytes, causing (1) disrupted mitochondrial integrity; (2) lower basal mitochondrial respiration; and (3) reduced ATP production and maximal mitochondrial respiration. The overexpression of carnitine palitoyltransferase, a protein that metabolizes LCACs in mitochondrial matrix, decreases LCACs levels improving the viability in cocultured neurons. Therefore, LCACs could represent either diagnostic or prognostic biomarkers, as well as a novel therapeutic target in IS [143].

During IS, the pathological increase in lysophosphatidic acid (LPA) represents another relevant lipid abnormality. Levels of this bioactive lipid, together with its producing enzyme autotaxin, are increased in the BBB in IS. These alterations disrupt BBB integrity, worsening stroke progression. The upregulation of autotaxin, and consequently of LPA, affects endothelial permeability, protein junctions and mitochondrial bioenergetics in mouse brain microvascular endothelial cells during I/R. The Seahorse approach revealed that LPA overexpression led to decreased OCR and higher levels of ECAR, causing energy production to shift to glycolysis. On the contrary, modifications in endothelial permeability are attenuated by autotaxin inhibitors in murine models. Since the decrease in LPA restores cell permeability and mitochondrial function, the inhibition of the autotaxin–LPA axis could be a therapeutic strategy to face stroke outcomes [144].

Recent studies have detected the involvement of different protein kinases belonging to the serine/threonine protein kinase family in stroke pathogenesis. For example, SGK, a member of the serine/threonine kinase family, is involved in inflammation, oxidative stress and ion transport. SGK1 is highly expressed in neurons of the cortex and hippocampus, which are highly susceptible to ischemia. The pharmacological inhibition of SGK1, through a specific inhibitor called GSK650394, enhances mitochondrial function in the hippocampus, resulting in improved basal respiration, ATP-linked respiration, proton leak-linked respiration and maximal respiration, leading to better neuronal survival. Moreover, in vivo studies showed relieved cortical hypoperfusion and neuroinflammation after SGK1 inhibition, resulting in better neurocognitive outcomes. Thus, SGK1 inhibition ameliorates neuronal injury due to its role in neuronal survival during brain ischemia [145].

Another member of serine/threonine kinase family is protein kinase N1 (PKN1), which is mostly expressed in the brain. Its deficiency positively regulates cerebral glycolytic flux and mitochondrial pyruvate-induced respiration, showing a protective effect in in vitro stroke models. Brain tissue of Pkn1^−/−^ mice shows higher levels of phosphofructokinase, an enzyme involved in glycolysis. Low levels of PKN1 are also linked to increased production of Glc-1,6-BP, a metabolite involved in mitochondrial pyruvate uptake, and the consequent increase in ATP levels. Moreover, the transcriptional pathways involved in the regulation of energy metabolism were upregulated in Pkn1^−/−^ brain slices. Therefore, inhibition of PKN1 could be a new therapeutic avenue to mitigate stroke progression [146].

Importantly, cells subjected to OGD/R insult redirect energy production through oxidative phosphorylation because glycolysis is unable to adequately satisfy the cellular ATP request [147]. Therefore, the metabolic switching from glycolysis to oxidative phosphorylation could represent a therapeutic approach against IS since it makes stem cells more resistant to ischemic insult. Mesenchymal stem cells grown under normal condition (nMSCs) and mesenchymal stem cells subjected to the metabolic switching paradigm (sMSCs) were assayed for OCR and ECAR. Importantly, the OCR/ECAR ratio was higher in sMSCs compared to nMSCs, showing a shift from glycolysis to oxidative phosphorylation in mitochondrial metabolism. The oxidative metabolism of switched cells made them more responsive to the therapeutic approach [148]. Moreover, when nMSCs and sMSCs were cocultured with neurons, sMSCs exhibited higher spare respiratory capacity and basal metabolic rate as compared to nMSCs, since they generated sMito. The transfer of sMito from sMSCs to OGD-exposed neurons improved cell viability, rescuing ischemia-dependent mitochondrial damage and demonstrating that sMito can play a crucial role in the viability improvement of OGD neurons [149].

Finally, the Seahorse approach recently revealed that mitochondrial ROS production during reperfusion is regulated through tissue succinate levels. Using comparative in vivo mouse models of I/R injury (brain, kidney, heart, liver) and ^13^C substrate tracing, it was found that succinate accumulates in all ischemic tissues and that the oxidation of succinate during reperfusion drives mitochondrial ROS accumulation and injury. Thus, the modulation of succinate metabolism during reperfusion may offer a viable approach to reduce I/R injury [10].

## 9. Therapeutic Perspectives Based on Recent Acquisitions on Metabolic Disturbances Involved in CVDs

As discussed in the above sections of this article, novel molecular mechanisms of disease are being uncovered through the metabolic approach (Figure 4).

The new knowledge may offer new opportunities for the development of therapeutic strategies, as illustrated in Table 1. Indeed, the exploration of mitochondrial-targeted drugs provides a burgeoning field of research with significant implications for CVDs. Future research is needed to understand the precise mechanisms by which these compounds exert their effects and to confirm these findings in patients.

## 10. Conclusions and Future Directions

Given that the heart is a high-energy-demanding organ, energy metabolism plays a central role in heart pathophysiology. Together with immune-metabolic alterations and endothelial dysfunction, the impairment of mitochondrial function accounts for the development of most CVDs, which are associated with an energetic deficit often generated by disturbances in one or more metabolic steps of ATP production. The multifaceted nature of mitochondrial dysfunction complicates our understanding of its contribution to several CVDs.

Nowadays, the availability of novel sophisticated tools through technological advancements allows us to study cellular bioenergetics and to achieve precise measurement of key metabolic parameters. We are becoming aware of the important contribution of metabolic alterations to the pathogenesis of common CVDs, such as those depending on mitochondrial dysfunction, altered mitochondrial dynamics, endothelial dysfunction and abnormalities of lipid and glucose metabolisms, although it is still unclear whether they may be either a cause or a consequence. The studies examined in this review collectively highlight the advancing knowledge on the metabolic alterations involved in CVDs and the tight link between bioenergetics and genetic, environmental, age and sex-related specific factors. Along with the need to develop new therapeutic strategies, as illustrated above, we underline the opportunity to identify new biomarkers of bioenergetic dysfunction (including lactate levels, TCA cycle metabolites and circulating mitochondrial DNA) to be used in the clinical setting. As a matter of fact, bioenergetic biomarkers can specifically aid in the early detection of heart diseases, risk stratification and monitoring treatment responses.

In conclusion, the future direction is represented by the development of a personalized approach that includes bioenergetic profiling to tailor interventions for individual patient needs, enhancing both efficacy and safety.

## Figures and Tables

**Figure 1 ijms-26-06791-f001:**
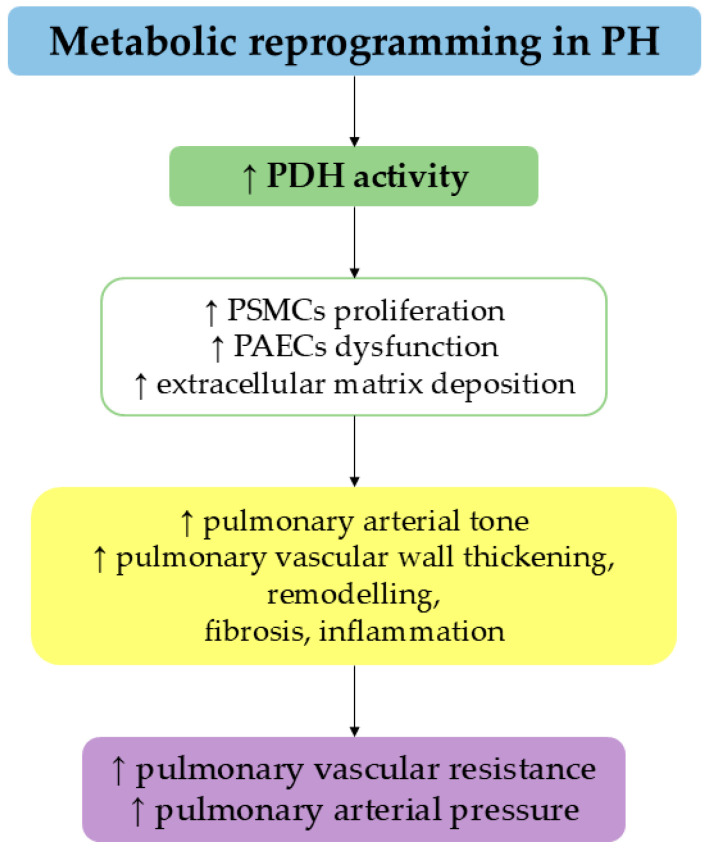
The impact of metabolic reprogramming on PDH activity modulates a series of events within the pulmonary vascular wall, ultimately leading to increased pulmonary vascular resistance and arterial pressure.

**Figure 2 ijms-26-06791-f002:**
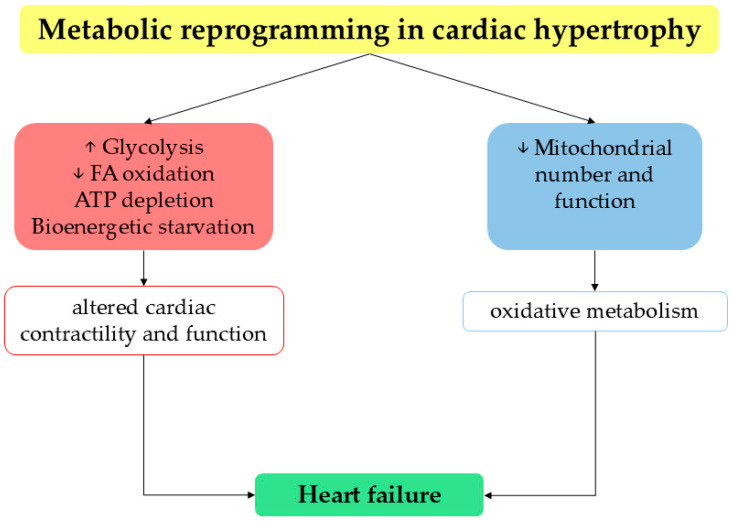
Schematic representation of key mechanisms underlying the development of cardiac hypertrophy and ultimately of HF. In fact, altered bioenergetics, along with the decreased number of mitochondria, compromise the efficiency of cardiac contractility.

**Figure 3 ijms-26-06791-f003:**
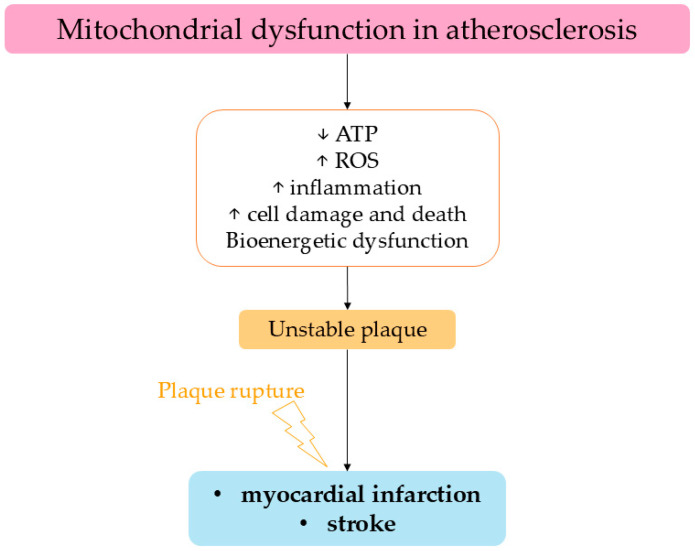
Bioenergetic dysfunction consequent to atherosclerosis favors plaque instability and the consequent plaque rupture causes known cardiovascular acute events.

**Figure 4 ijms-26-06791-f004:**
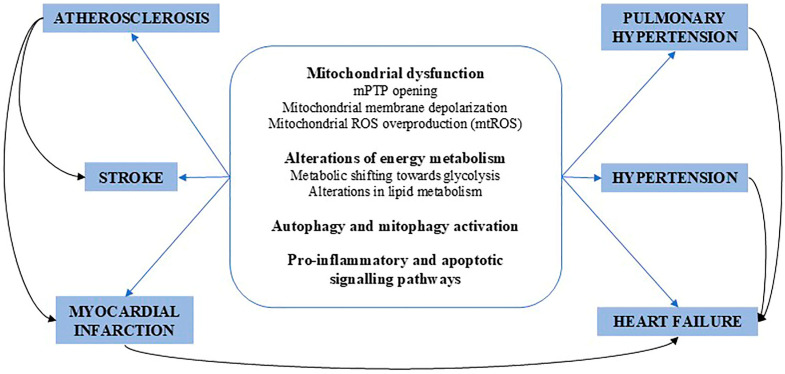
Schematic representation of the key metabolic abnormalities shared by common CVDs.

**Table 1 ijms-26-06791-t001:** List of relevant molecules involved in metabolic disturbances underlying CVDs and their potential therapeutic implications.

Molecule	Biological Activity	Suitable CVD to Treat
**CALCA**	It inhibits sFLT-1	**Preeclampsia**
**FPER-1**	It regulates PASMCs proliferation through inhibition of PI2K/Akt axis	
**AIF**	Mitochondrial oxidoreductase	**Pulmonary** **Hypertension**
**FAS**	Its inhibition improves mitochondrial function, mitochondrial respiratory capacity and increases ATP level reducing oxidative stress	**Pulmonary Hypertension**
**ENO1**	Its inhibition improves endothelial and mitochondrial dysfunction in PAECs through PI3K-Akt-mTOR pathway, and it improves right ventricular function in vivo	
**CPT-I**	Its inhibition regulates mitochondrial FA oxidation and prevents LV dysfunction	**Heart Failure**
	It increases mitochondrial respiration and biogenesis in vitro; it preserves cardiac diastolic function in vivo	
**Salsolinol**	It improves mitochondrial respiratory function and energy metabolism by inhibiting the excessive activation of a specific mitochondrial calcium uniporter	
**MitoTEMPO**	It restores mitochondrial function and reduces mtROS	**Atherosclerosis**
**KLF7**	It reduces atherosclerosis by limiting glucose metabolic reprogramming through HDAC4 activation	
**NPM1**	Its inhibition improves myocardial damage and promotes cardiac remodeling post infarction	**Myocardial** **Infarction**
**Muscone**	It exerts anti-inflammatory activity, anti-apoptotic property and promotes glycolytic flux	
**DMF**	It exerts anti-inflammatory effects and improves oxidative phosphorylation	
**FtMt**	It improves mitochondrial function, protects from neuronal apoptosis and OGD/R energy stress	**Stroke**
**Autotaxin-LPA**	Its inhibition restores cell permeability and mitochondrial function	
**SGK1**	Its inhibition improves mitochondrial function, basal respiration and neuronal survival	
**PKN1**	Its inhibition exerts positive regulation of cerebral glycolytic flux and of mitochondrial respiration and it increases ATP levels.

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
