# Peer review of "Metabolic Disturbances Involved in Cardiovascular Diseases: The Role of Mitochondrial Dysfunction, Altered Bioenergetics and Oxidative Stress"

_ijms, 2025, doi:10.3390/ijms26146791_

Round 1
Reviewer 1 Report
Comments and Suggestions for Authors
Recommendations:
- High percent match, please reduce it!
- The manuscript is rather monotonous, please add some descriptive figures to illustrate the impact on cardiovascular disease.
- The authors could add JAK2 kinase as a very important molecule implicated in cardiovascular disease; see this: https://doi.org/10.3390/cimb46080496
- Also the impact of short-chain fatty acids (SCFAs) on cardiovascular disease : https://doi.org/10.3390/jcm14082678
Overall, the manuscript is very hard to read due to its monotonous structure, a review like this that assesses issues that are mainstream now, should give more impact to attract readers, also I did not see any novelty element in this review.
Comments on the Quality of English LanguageMinor english revision!
Author Response
Please find the attachment below.

Reviewer 2 Report
Comments and Suggestions for Authors
Dear Authors,
Respectfully, I have shared my comments and suggestions, which I hope will contribute to enhancing the clarity, coherence, and scientific rigor of your manuscript.
Title
Comment: I suggest adjusting the title to more accurately reflect the central focus of the manuscript, highlighting mitochondrial dysfunction, bioenergetics, and oxidative stress, which are clearly the main axes developed throughout the text.
Introduction
Lines 33-37. Please add references.
Comment: The text mentions glucose as the main energy source, but it is important to point out that in cardiac and vascular physiology, fatty acids, ketone bodies, lactate, and amino acids also play fundamental roles in energy generation.
Comment: The introduction does not clearly state that dysfunction of the mitochondrial respiratory chain leads not only to reduced ATP production but also to increased ROS generation, mitochondrial DNA damage, and activation of inflammatory and apoptotic pathways.
Comment: While the introduction presents the basic concepts of metabolism well, it would be very enriching to include an integrated view of different cell types (cardiomyocytes, endothelial cells, vascular smooth muscle cells, and neurons), highlighting that bioenergetic dysfunction is a common denominator across cardiovascular, cerebrovascular, and renal diseases. Including a figure that illustrates the common axes, such as ATP generation, ROS overproduction, oxidative stress, mPTP opening, mitochondrial membrane potential loss, calcium dysfunction, mitochondrial biogenesis, and mitophagy.
Comment: The transition to the discussion of CVDs is somewhat abrupt at line 68. I suggest adding a paragraph that builds this connection, explaining why energy metabolism is particularly critical in these pathological contexts.
Comment: The text introduces the abbreviations OCR (Oxygen Consumption Rate) and ECAR (Extracellular Acidification Rate) without defining them.
Comment: I recommend that the authors clearly state the specific objective of the review at the end of the introduction.
Hypertension
Comment: This section could be greatly enriched by also discussing essential hypertension, which accounts for the majority of cases in clinical practice.
Comment: Although the role of angiotensin II in endothelial dysfunction is mentioned, it would be important to further elaborate on how this dysfunction directly impacts mitochondrial metabolism in both endothelial cells and vascular smooth muscle cells, which are key regulators of blood pressure.
Comment. The section is heavily focused on renal and endothelial aspects but lacks a more comprehensive discussion on how systemic metabolic dysfunction, including obesity, insulin resistance, and dyslipidemia, contributes to the development and worsening of hypertension, particularly through inflammatory and immunometabolic mechanisms.
Comment. The section briefly mentions microRNAs, but it does not specify which are the key regulators implicated in hypertension.
Comment. The section relies predominantly on animal and cellular models. It would be important to discuss whether these findings have been validated in human studies or if there are ongoing therapeutic developments based on these mechanisms.
Comment. Hypertension shows well-documented differences related to biological sex and aging, both of which are intimately linked to mitochondrial dysfunction and increased oxidative stress. Including this perspective would broaden the discussion and align it with current evidence.
Pulmonary Hypertension
Comment: This section is highly focused on mitochondrial metabolism in PASMCs and PAECs but lacks a clear discussion on how these cellular alterations directly translate into clinical outcomes, such as increased pulmonary vascular resistance, right ventricular dysfunction, and disease progression.
Heart Failure
Comment: Although the section briefly mentions the involvement of CCR2+ monocytes and macrophages, it does not delve into how inflammatory metabolism contributes to the progression of heart failure.
Comment: Endothelial bioenergetics, which plays a critical role in heart failure, especially in HFpEF, is given very little attention..
Atherosclerosis
Comment: This section would be strengthened by establishing a clearer link between the cellular and metabolic alterations discussed and the hemodynamic and structural events that lead to plaque rupture, thrombosis, myocardial infarction, stroke, and sudden death.
Comment: The integration of bioenergetic mechanisms with the classical processes of atherogenesis is missing. These include endothelial dysfunction, LDL oxidation, immune cell recruitment, foam cell formation, and extracellular matrix remodeling.
Comment: Although the section briefly mentions triglycerides, LDL, and glucose, it does not adequately discuss how metabolic conditions such as obesity, insulin resistance, diabetes, and metabolic syndrome directly impact mitochondrial metabolism, oxidative stress, and consequently, plaque formation and progression.
Myocardial Infarction
Comment: Despite the focus on cardiomyocytes and macrophages, the section does not address endothelial bioenergetic dysfunction, which is critical in reperfusion injury, microthrombus formation, and impaired coronary perfusion post-infarction.
Comment: The section discusses the differentiation of Ly6C^high and Ly6C^low macrophages but does not explore how cellular metabolism regulates this immune plasticity.
Comment: There is no discussion regarding mitochondrial dynamics, specifically fusion (MFN1, MFN2, OPA1), fission (DRP1), and mitophagy, which are well-established as key mechanisms in either protecting against or exacerbating myocardial damage after infarction.
Comment: I recommend that the authors consider discussing how key clinical factors, such as age, biological sex, the presence of comorbidities (e.g., diabetes, obesity), and environmental factors, impact bioenergetics and post-infarction outcomes.
Stroke
Comment: This section provides a very good analysis of cellular bioenergetic mechanisms in stroke, focusing on neurons, astrocytes, and partially on endothelial cells. However, it lacks a clear explanation of how these cellular alterations translate into major pathophysiological events, such as cerebral edema, blood-brain barrier dysfunction, microthrombosis, extension of the ischemic lesion, and progression of neurological deficits.
Conclusion
Comment: Including a figure and a table in the conclusion section is not common practice in scientific manuscripts. The conclusion should serve as a space to synthesize the main findings, provide critical reflection, discuss limitations, and outline future directions.
Comment: I suggest that the conclusion incorporates a more comprehensive view, integrating mitochondrial dynamics (fusion, fission, mitophagy), immunometabolism, and endothelial dysfunction as central components in the pathophysiology of CVDs.
Comment: I recommend that the authors outline clear future research directions, including the development of clinical trials targeting mitochondrial therapies, the identification of bioenergetic dysfunction biomarkers, and investigations into how genetic, environmental, age-related, and sex-specific factors modulate mitochondrial function and cardiovascular risk.
Best regards.
Author Response
Please find the attachment below.

Round 2
Reviewer 1 Report
Comments and Suggestions for Authors
Congratulations, for addressing all comments!
Reviewer 2 Report
Comments and Suggestions for Authors
Dear Authors,
Thank you for your careful and thorough revisions.
I reviewed the updated manuscript and your responses, and I’m pleased to see that all previous comments were addressed with clarity and scientific rigor. The improvements have strengthened the work considerably.
Best regards.